# Enhancing Role of Guiding Signs Setting in Metro Stations with Incorporation of Microscopic Behavior of Pedestrians

**Bin Lei [1], Jinliang Xu [2,\*], Menghui Li [3,\*], Haoru Li [2], Jin Li [4], Zhen Cao [1], Yarui Hao [1] and Yuan Zhang [1]**

[1] School of Civil Engineering, Xi'an University of Architecture and Technology, Xi'an 710055, China; leibin@xauat.edu.cn (B.L.); professorcaozhen@163.com (Z.C.); haoyarui@xauat.edu.cn (Y.H.); zhangyuan19@xauat.edu.cn (Y.Z.)
[2] College of Highway Engineering, Chang'an University, Xi'an 710064, China; 2017021045@chd.edu.cn
[3] China Harbour Engineering Company Limited, Beijing 100027, China
[4] Department of Highway and Architecture, Shandong Transport Vocational College, Weifang 261206, China; lijin891103@outlook.com
\* Correspondence: xujinliang@chd.edu.cn (J.X.); menghui_li@outlook.com (M.L.)

**Abstract:** In the metro operation environment, guiding signs provide direction and route conversion instructions to pedestrians. In metro stations with massive passenger flow, the rationality of sign setting would exert distinct effects on the efficiency of passenger flow. Currently, most studies on guiding signs focus on architecture, aesthetics and simulation. However, perspectives from humanization of pedestrian guidance signs such as pedestrian behavior needs and pedestrian cognition were seldom proposed. In this paper, the microscopic behavior characteristics data of pedestrians at different positions in typical metro stations were collected through pedestrian tracking experiments. After analyzing the characteristics of pedestrians' microscopic behavior in metro stations, otherness of walking speed was found out among pedestrians in different types of passageways. The walking speed of pedestrians in closed-type passageways is higher than other types. Moreover, pedestrian speed at the stairs adjacent to the platform is higher than that at the stairs not adjacent to the platform. With the increase of crowd density, the change of walking speed of pedestrians can be represented by a unimodal curve. Finally, the key points of optimal setting of guiding signs in different regions and different periods were obtained according to the result analysis of the experiment. The research results of this paper can provide theoretical support and technical guidance for the optimal establishment of pedestrian guiding signs in metro stations with massive passenger flow.

**Keywords:** microscopic behavior; guiding signs; metro station; massive passenger flow

## 1. Introduction

As an important form of rail transit, metro has become the main mode of public transit in urban transportation. In China, metro stations provide service to a large number of passengers to maintain normal running of cities in a routine day. Therefore, its operation efficiency will affect the whole urban public transportation system. Effective setting of guiding signs would improve operation efficiency of the public transport system, or it might hinder the normal operation of metro stations, and further lead to serious traffic accidents. Therefore, guiding signs play an important role in passenger flow administration, attracting worldwide attention on their setting method. A stream of studies have been proceeded to emphasize the role of guiding signs in metro stations [1–5]. Among this research, Ichiro summarized the whole process of pedestrian guidance system design of Yokohama metro station in

Japan, and studied the standardization of content and elements of guiding signs [6]. Filippidis et al. proposed the concept of VCA (visibility field), simulated the interaction between pedestrian and sign system, and analyzed the impact of exit sign on evacuation process by using buildingEXODUS software [7]. Nassar simulated the three-dimensional interaction of pedestrians, and considered the perception of pedestrians of space in the study to explore the optimal location of guiding sign setting [8].

In recent years, research has realized the importance of pedestrian psychology in the study of guiding sign setting, aiming to consider the setting of guiding signs from a people-oriented perspective [9–12], such as interactions between pedestrians and surrounding environments [13], and pedestrian recognition of signs [14]. The reasonability and completeness of guiding sign setting in metro stations can show evident effects on efficiency of pedestrian flow, reflecting the level of service from a systematic level. Therefore, in assurance of the robust and efficient operation of metro system, the necessity of behavioral characteristics analysis of pedestrians should be emphasized to provide practical reference for the optimal setting of guiding signs. Through experiments in real scenarios, Vilar stated that guiding signs have greater guidance effect on daily pedestrians than on emergency situations. The reason for this phenomenon was pedestrians' path-finding behavior is more rational in daily situations [15]. Wang believed that the visibility of signs is related to the attractiveness of the target, the distance between pedestrians and the target, and the occupancy rate of the target in vision of pedestrians [16].

Microscopic behavior of pedestrians is a good reflection of pedestrian psychology, it has been studied by researchers to better understand pedestrian psychology. Zhao studied the behavioral characteristics of pedestrian flow speed on location where metro station is connected with shopping mall, understanding relationships between moving speed and personal information, such as age, gender, luggage, shoes, and accompanying [12]. Chen proposed a multiagent-based model for pedestrian simulation in subway stations, taking staged pedestrian behaviors into account, which works well for simulating real pedestrian behaviors in metro stations [17].

However, the importance of linking reasonable setting of guiding sign and behavioral characteristics of pedestrians was not realized by researchers until recently [18]. The main stream of this research studied behavioral characteristics of pedestrians under context of assumed evacuation scenarios in metro stations [19–22], while for administrators of metro operation and citizens using metro services, the key point should be assurance of efficient pedestrian flow in metro stations under normal situations for most of the time. Moreover, the psychology and behavioral intention of passengers in metro stations are quite different under context of normal operation compared to emergency evacuation [23–25]. Therefore, the viability and applicability of research results under assumed emergency evacuation situations cannot be directly applied to massive passenger flow under normal situations, evoking the necessity of research on guiding sign setting for metro users under normal operation. To the best of the authors' knowledge, no relevant study has been performed to date to explore the setting of guiding signs from perspectives of practical pedestrian behavior needs and pedestrian cognition. Actual needs of pedestrians reflected by the micro-behavior of pedestrians in the actual environment were ignored, which would limit practical application.

The setting of guiding signs should be in accordance with behavioral habits of users to maximize their function on pedestrian guidance. In this paper, the microscopic behavior characteristics data of pedestrians at different positions in typical metro stations were collected through pedestrian tracking experiments. After analyzing the characteristics of pedestrians' microscopic behavior in metro stations, otherness of walking speed was found out among pedestrians in different types of passageways; the walking speed of pedestrians in closed-type passageways is higher than other types. Moreover, the pedestrian speed at the stairs adjacent to the platform is higher than that at the stairs not adjacent to the platform. With the increase of crowd density, the change of walking speed of pedestrians can be represented by a unimodal curve. Finally, the key points of optimal setting of guiding signs in different regions and different periods were obtained according to the result analysis of the experiments.

The study might fill the gap that is a lack of relevant study focused on improvement of efficiency of pedestrian flow in metro stations under normal situations, from the perspective of pedestrian psychology and microscopic behavior. Moreover, the research results of this paper can provide theoretical support and technical guidance for the optimal establishment of pedestrian guiding signs in metro stations with massive passenger flow.

The remainder of the paper has been organized as follows. Section 2 introduces the details of the field experiments. The results are analyzed in Section 3. Section 4 provides the setting methods of guiding signs and the verification of proposed setting method. The research is summarized in Section 5.

## 2. Field Experiment on Microscopic Behavior of Pedestrians

### 2.1. Purpose

Microscopic behavior of pedestrians refers to characteristics of individual behavior under certain circumstances or facilities. By on-the-spot tracking of metro users, the behavior data of pedestrians can be obtained. The data can be used to analyze the behavior characteristics of pedestrians in different areas. After this, the guidance information needs of different areas can be obtained by further analysis, so as to improve the identification setting.

Currently, no field data concerning pedestrian behavioral needs in China on guiding signs are available. Therefore, field experiments were conducted to obtain valid data of pedestrian behavior and cognition, aiming to understand the needs of pedestrians on guiding signs.

### 2.2. Experimental Design

#### 2.2.1. Experimental Object

The selection of experimental object should approximately meet the real age and gender distribution of pedestrians in the station to reflect the actual characteristics of the station. Therefore, the object covered pedestrian samples of different ages, genders and travel purposes.

#### 2.2.2. Location of Experiment

Xiaozhai station is a typical transfer metro station in Xi'an city, and the service objects of this station are diverse, which can fully reflect the behavior characteristics of all kinds of pedestrians and effectively ensure the scientificity and validity of the experimental results. Therefore, Xiaozhai station was selected as the experimental research site of pedestrian microscopic behavior tracking.

#### 2.2.3. Duration of Experiment

Based on the statistical analysis of the passenger traffic volume in and out of the station at different time periods of a week, combined with the travel purpose and composition of pedestrians, the collection of data was divided into three parts: Weekend peak hour data was collected from 12:00 to 20:00 on Sunday, weekday peak hour data was collected from 18:00 to 19:00 on weekdays, and weekday flat hour data was collected from 9:00 to 11:00 on weekdays.

#### 2.2.4. Experimental Procedure

Step 1: Familiarizing with the space structure inside the station, number of tracking control points inside the station and previews of tracking process.

Step 2: Random assignment of the experimenters to wait for the metro outside the waiting line at the platform floor.

Step 3: Each experimenter randomly selects a get-off passenger as the tracking target after arrival of the metro, recording the starting time and starting position without affecting their normal behavior.

Step 4: Draw the walking route of the pedestrian on the thumbnail of the station structure according to the actual walking situation of pedestrians during the tracking process, and record the stopping point and stopping time.

Step 5: After this, the experimenter should invite the pedestrian to complete a questionnaire containing his/her basic information, and inquire and record the reasons for the pathfinding behavior. The questionnaire should be finished on the premise voluntarily.

Step 6: Adjust the tracking object selection according to the requirement of sample richness and composition structure, and repeat Step 2–Step 5 until required sample number are collected.

The illustration of path of transfer path of one sample is shown in Figure 1. The order of walking is from 1 to 3.

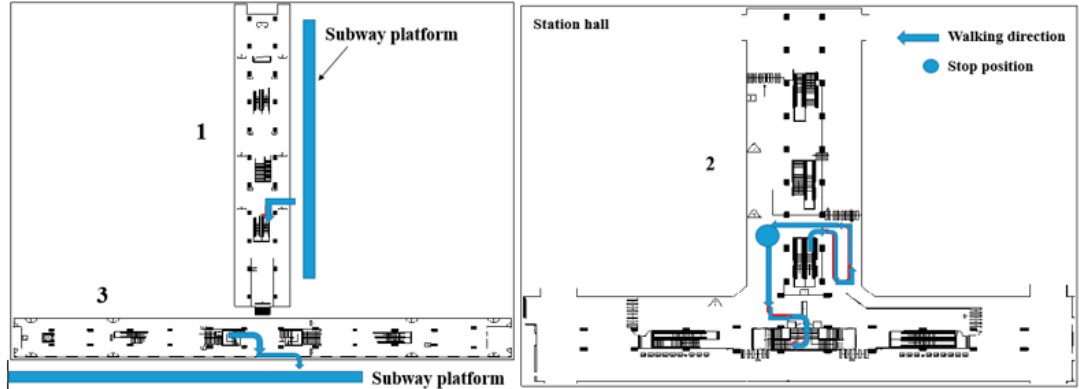

**Figure 1.** Illustration of transfer path of pedestrian.

## 3. Analysis of Observed Pedestrians' Microscopic Behavior Characteristics

### 3.1. Analysis of General Tendency

A part of the statistics of samples of line three in Xiaozhai station is listed in Table 1.

**Table 1.** Partial statistics of samples of line three in Xiaozhai station (outbound direction).

| Starting Time | Age | Gender | Level of Education | Walking Time at Platform | Walking Time in Station Hall | Duration of Stagnation (s) | Reasons |
|---|---|---|---|---|---|---|---|
| 8:25 | 25 | male | bachelor | 37 | 98 | | |
| 8:23 | 26 | male | bachelor | 38 | 47 | | |
| 9:40 | 70 | female | high | 22 | 75 | 3 | |
| 9:36 | 74 | male | primary | 9 | 115 | 5 | |
| 10:30 | 24 | female | bachelor | 29 | 67 | | |
| 10:05 | 35 | male | college | 36 | 70 | | |
| 10:05 | 27 | female | college | 11 | 63 | | |
| 10:20 | 65 | male | primary | 23 | 216 | 5 | Unclear information |
| 15:55 | 17 | male | high | 10 | 59 | | |
| 16:25 | 22 | female | college | 7 | 61 | | |
| 16:07 | 31 | male | bachelor | 12 | 190 | 127 | Unclear information |
| 15:40 | 32 | female | bachelor | 14 | 33 | 5 | Unclear information |
| 15:57 | 32 | female | college | 8 | 77 | | |
| 16:40 | 23 | female | college | 31 | 171 | 120 | |
| 16:00 | 23 | female | bachelor | 27 | 125 | 20 | Unclear information |
| 16:24 | 19 | female | bachelor | 59 | 92 | 30 | |
| 16:54 | 18 | female | high | 38 | 47 | 15 | Unclear information |
| 16:54 | 17 | female | high | 38 | 48 | 15 | Unclear information |
| 16:18 | 28 | female | bachelor | 13 | 65 | 20 | Unclear information |
| 18:19 | 23 | female | bachelor | 8 | 119 | 53 | Unclear information |

To highlight the microscopic behavior characteristics of pedestrians under massive passenger flow, taking the pedestrian behavior characteristics during peak hours and flat hours as comparison. Taking the data analysis of the pedestrian tracking survey of line three and line two of Xiaozhai metro station

as an example, the walking time of different samples in flat hours and peak hours was calculated and sorted according to the time consumption. The recorded walking time was the time after deducting the behavior unrelated to wayfinding performance (phone, chat with friends, etc.). As shown in Figure 2, the walking time of different pedestrians during flat hours is relatively balanced, while the walking time of different pedestrians during peak hours varies greatly. The appearance of the intersection in the figure indicates that walking speed during crowded peak hours even exceed the speed during flat hours. The reason for this phenomenon might be explained as, during flat hours, (a) the pedestrian can move forward at their corresponding desired speed due to the low crowd density in the station, and (b) they receive relative low disturbance from the surrounding crowd. However, pedestrians would receive much higher disturbance from the surrounding crowd because of high crowd density in the station during peak hours. Some of them would surpass others due to different travel purposes, time value perception and emotions that want to leave the crowded environment as soon as possible, etc. Therefore, some pedestrians' walking time during peak hours is less than that during flat hours.

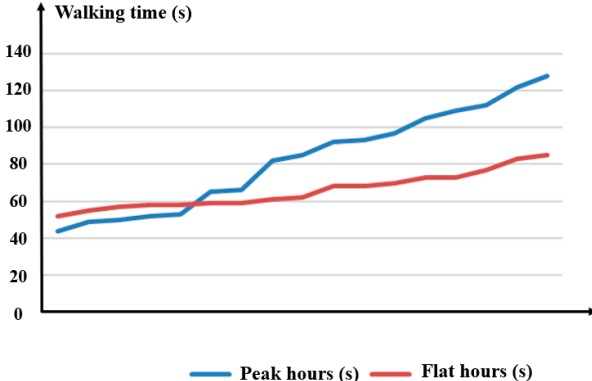

**Figure 2.** Comparison chart of walking time during peak hours and off-peak hours.

The relationship between walking time and crowd density is shown in Figure 3. In general, the walking time showed a trend of decreasing first and then increasing with crowd density. As the pedestrians can walk at their desired speed, obvious individual characteristics and short walking time were observed among pedestrians under low crowd density. With increase of crowd density, pedestrians in front will speed up to avoid being surpassed under the pressure of the pedestrians behind, and pedestrians behind will follow up quickly and surpass the pedestrians in front if possible. At this time, the walking speed of the pedestrians increases significantly and the walking time is the shortest. When the crowd density reaches its limit, the walking speed of pedestrians decreases obviously due to space restriction, at which time the individual characteristics of pedestrians faded and passenger flow characteristics magnified.

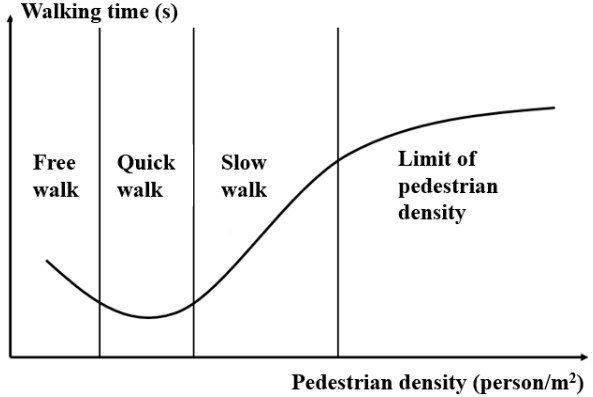

**Figure 3.** Correlation between density and walking time.

According the statistics of pedestrian behavior recorded by experimenters, the duration of stay at different locations of the station are illustrated in Figure 4. Detailed analysis of micro-behavior characteristics of pedestrians is provided in the next section.

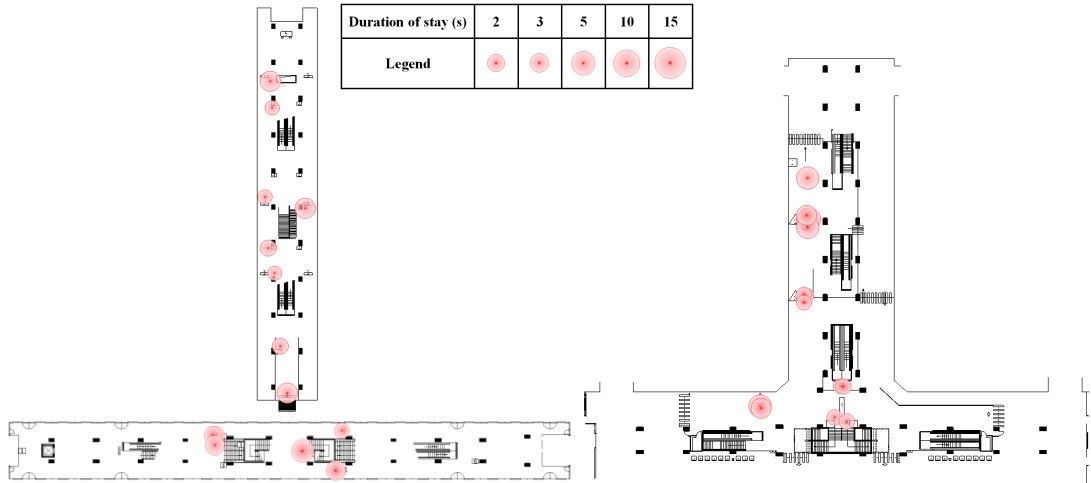

**Figure 4.** Illustration of duration of stay at different locations.

### 3.2. Detailed Analysis of Micro-Behavior Characteristics of Pedestrians

#### 3.2.1. Pedestrian Behavior Characteristics at Horizontal Passageway

Horizontal passageway can be classified into open passageway, enclosed passageway, and semi-enclosed passageway according to different building forms. Normally, the open passageway can provide a better transition space, thus the passenger flow is concentrated, the pedestrian density is high, and the pedestrian walking speed is fast. It is observed that in open passageways without separation, the pedestrians often choose a shorter path rather than follow the principle of waling on the right to shorten walking time under low crowd density. Therefore, the interweaving of passenger flow in the self-organizing behavior of the crowd is serious under peak hours. Enclosed passageway connects different functional spaces directly through independent channels, such as single outbound channels, transfer channels, or two-way transfer channels with dividers. In the transfer channel or exit channel, the direction is easy to identify, but it is not convenient to switch from other functional spaces, and the walking distance is long. According to the experimental data, the speed of pedestrians in the enclosed passageways is higher than that of other positions. The main reasons include: (a) Pedestrians have a clear direction rather than searching state in the enclosed passage, and (b) the psychological insecurity of pedestrians increases in a closed space, and pedestrians move faster to leave this environment as soon as possible. Semi-enclosed passageway is usually connected with the underground commercial space of the building, and it can evacuate pedestrians to the ground or other functional spaces through the interior of the building. However, pedestrians are easily confused and the interweaving of passenger flow is serious.

According to the statistics, the walking speed of men is about 10% higher than that of women in the same environment, and the walking speed is greatly affected by the passageway environment. The enclosed passageway has the fastest walking speed, followed by the open passageway and the semi-enclosed passageway.

#### 3.2.2. Pedestrian Behavior Characteristics at Stairs

In metro stations, stairs are of great importance for vertical transportation, which can be categorized as walking stairs, escalators and elevators. The escalators and elevators provide invariable speed and fixed traffic volume, which would limit the pedestrians' walking behavior under large passenger

flow. Since the pedestrian behavior characteristics on escalators or elevators are fixed, only pedestrian behavior characteristics at walking stairs are analyzed. The behavior characteristics of pedestrians at the walking stairs are different from those in the transfer channel or exit channel. The behavior of pedestrians on the stairs is influenced by the depth, height and inclination of the stairs, as well as the physiological factors and time requirements of pedestrians. According to statistics, whether the stairs are adjacent to the platform or not is the main reason affecting the walking speed of pedestrians. The walking speed up and down the stairs that directly connected with the platform is about 20% higher than those not connecting the platform. In addition, it was found that pedestrians walk near the handrail when walking up and down stairs.

### 3.2.3. Pedestrian Behavior Characteristics at Station Hall

The passenger flow lines in the station hall are complex because of multiple conflict points and confluence points, as shown in Figure 5. It is found that in the station hall without iron horse isolation, pedestrians with the same travel purpose have self-organization behavior, and passenger flow is automatically separated under the guidance of guiding signs. The phenomenon of self-organization of pedestrians is owing to the herding behavior, reflected by following the pedestrians in front. With the increase of the conflict points of pedestrian streamline and the increase of the passenger flow during rush hours, the pedestrian speed would decrease dramatically.

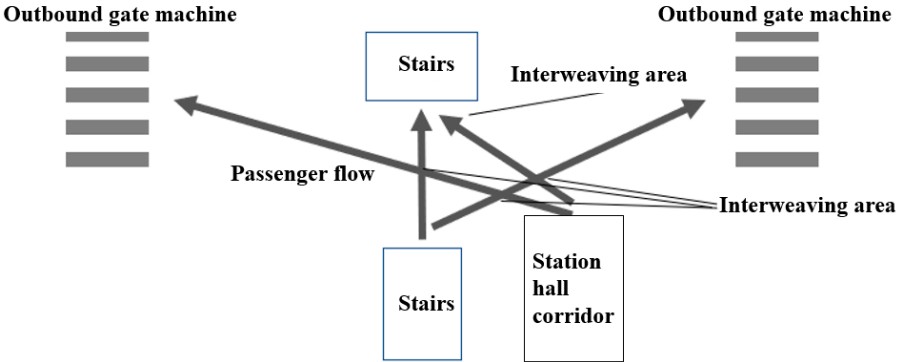

**Figure 5.** Illustration of interweave of pedestrian flow at the station hall.

### 3.2.4. Behavior Characteristics of Alighting Pedestrians at the Platform

The passenger flow at the platform is characterized by high concentration, uneven spatial and temporal distribution, and multiple conflict points. Congestion phenomenon was observed at gathering and distribution locations by alighting and transferring during rush hours, resulting in the reduction of pedestrian walking speed. Further, they would flock to the stairs within a short period, resulting in pulsed pedestrian congestion. Therefore, a fan-shaped accumulation is formed at the front of the platform stairs. In addition, pedestrians who are not familiar with the station are apt to look around and stop after alighting.

### 3.2.5. Characteristics of Pedestrian Behavior at Outbound Gate Machine Area

By comparing the pedestrian behavior in each area of the station, it is found that the pedestrian speed is lowest when passing through the gate machine area. walking space is limited and walking speed decreases dramatically in this area. In addition, the stop duration in this area is various according to individual conditions (age, degree of familiarity with gate machine, whether carrying luggage, etc.). The changing process of walking speed around the outbound gate machine area can be divided to normal walking area, walking speed buffer area (slow down), walking speed adjustment area (speed up), as shown in Figure 6.

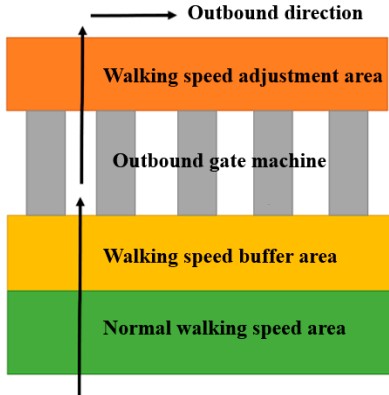

**Figure 6.** Illustration of speed change around gate area.

## 4. Pedestrian Behavior Characteristics-Based Guiding Sign Setting

### 4.1. Principles of Setting of Guiding Sign

The location of pedestrians affects pedestrians' demand for guidance information. Pedestrians conduct activities in different spatial areas of the station that are suitable for their corresponding functions. Therefore, appropriate guiding signs should be settled to provide pedestrians with required information. According to the analysis results of field research, combined with the current situation of metro and the research status of guiding sign setting, the sign setting in metro stations should follow the principles of continuity, eye-catching, readability, moderate amount of information, integrity and flexibility.

### 4.2. Guiding Sign Setting at Different Locations

4.2.1. Guiding Sign Settings in Horizontal Passageways

The pedestrian passageway in metro stations can be divided into one-way passageway and two-way passageway according to the walking direction of pedestrians. The following analysis is carried out on the establishment methods of the pedestrian guide signs in these two types of passageways.

- One-way passageway

Pedestrians walk in the same direction in a one-way passageway with explicit purpose, and the passenger flow is easy to organize and evacuate. In a one-way passageway, the setting of pedestrian guiding signs is conducive to relieve the sense of urgency of pedestrians. Considering the space limitation in one-way passageway, the type of guiding sign can be selected as suspension and wallpaper. The main information provided should be the transfer information to eliminate the unsureness psychology of pedestrians. The suspended guiding sign is easier to be recognized by pedestrians under massive passenger flow scenario without sight blocking. When setting the same guidance information, continuity and repeatability of information should be considered to impress the pedestrians. The setting of space interval can refer to former research results of short-term memory forgetting rule of pedestrians, and reset the sign within 20 s walking distance.

- Two-way channel

Two-way passageway is prone to interweaving and conflicts. The layout of pedestrian guiding signs can affect the walking route and walking efficiency of pedestrians walking in opposite directions by separating paths. In addition, according to the current environmental situation, iron horses can be reasonably set up to divide pedestrians and restrict them on their own ways. Continuity of guiding signs can be strengthened to avoid confusion and stagnation of pedestrians. The suspended form of guiding signs is preferred on prerequisite of continuity, matching with banded guiding signs to enable continuity of passenger flow, as shown in Figure 7.

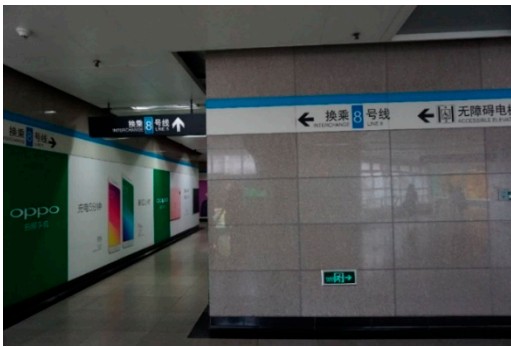

**Figure 7.** Continuous ribbon guidance sign and suspension guidance sign in the transfer channel (Shanghai).

### 4.2.2. Guiding Sign Settings at Stairs

Pedestrians have different demands for information at stairs in different locations, resulting in different psychological and behavioral characteristics of pedestrians. The following analysis is carried out on the setting of guiding signs at the stairs connecting different space positions.

- Stairway connecting platform

According to the field experiment, pedestrians will improve their walking speed out of the mentality of avoiding a long queue. For a two-way island platform, pedestrians that are unfamiliar with the station need to judge the direction of the metro first. Therefore, the pedestrian guiding sign at the location of the stairs should provide pedestrians the direction off the metro, so as to eliminate the urgent and uncertain emotions of pedestrians. Here, the sign form is recommended as suspension, and the content of the sign should include the name of the next stop or the final destination for corresponding direction, the number or color of the line and direction guidance. Guiding signs should meet the requirements of eye-catching and quickly identified by pedestrians to avoid the accumulation of pedestrians because of unclear information.

In exit direction, alighting passengers prefer the nearest stair to exit or transfer. A guidance sign containing exit information, transfer information, direction indication, and emergency exit indication should be set on this streamline. Suspension is preferred in the setting form, as shown in Figure 8.

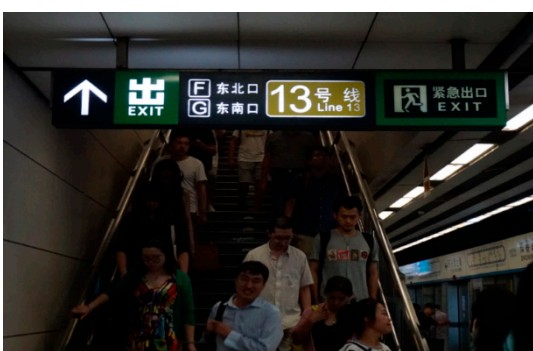

**Figure 8.** Setting of stair hanging guide sign at platform (Beijing).

- Stairway connecting passageway and station hall

For stairway connecting passageway and station hall, the guiding sign at the stairs is responsible for informing pedestrians of exit information, transfer information or station information. To continuously and rapidly guide pedestrians, the guiding signs are usually of suspension or wallpaper type, which meet the requirements of visibility and continuity. The content should include line name, station name, exit number, and other information. The information of transfer line should be included in the

guiding sign at stairs for transfer. The setting form of the guiding sign at the stairway connecting the access to the station is shown in Figure 9.

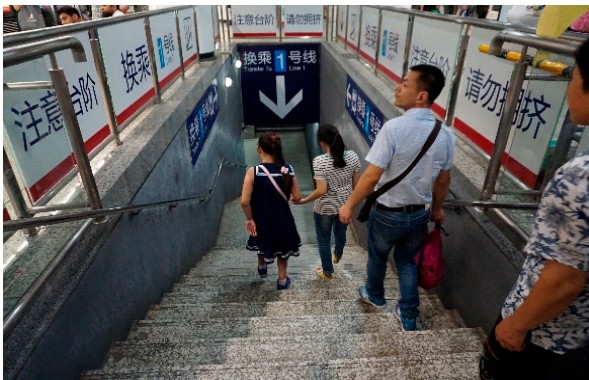

**Figure 9.** Setting of guiding signs at the inbound and outbound connector stairway of the station (Nanjing).

### 4.2.3. Guiding Sign Setting at Gate Machine Area

Gate machines in metro stations are divided into two forms: Inbound gate machines and outbound gate machines. According to the analysis of pedestrian behavior characteristics and needs in the gate area mentioned above, guiding signs at gate areas can be set according to the following methods:

- Inbound gate machine area

According to field experiments of pedestrian behavior characteristics at inbound gate area, it is found that pedestrian behavior is affected by both their individual characteristics and surroundings. After entering the station by swiping the card, pedestrians familiar with the station can quickly choose the path to metro, while pedestrians unfamiliar with the station would have behavior of stagnation and inquiry. To meet the time and safety requirement of commuting and evacuation, information such as the direction of the ride line should be informed to pedestrians in advance in front of the entrance gate. The content should cover information of driving direction, line number or color, etc., as shown in Figure 10a.

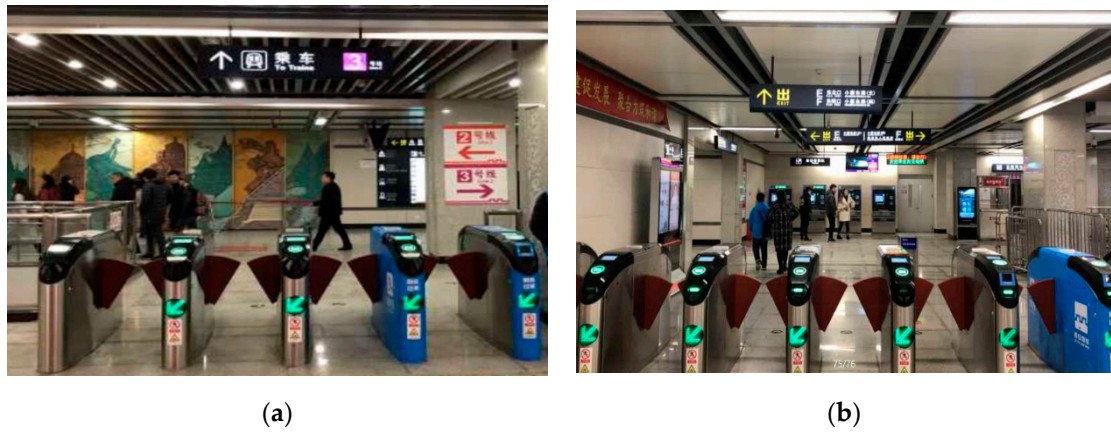

(**a**)　　　　　　　　　　　　　　　　　　　　　　　　　　　　(**b**)

**Figure 10.** Setting of hanging guide signs at brake area (Xi'an): (**a**) Entrance; (**b**) exit.

- Outbound gate machine area

Pedestrians have the behavior characteristics of slowing down in front of the gate. Long queues can form during rush hours. The main contributors of deceleration or stagnation are passengers who are not familiar with the station. Aiming to meet the needs of pedestrians for information after exit,

the information of exit should be informed in advance to assist pedestrians to choose the right exit. The content should cover exit information, ground information outside the station, transfer information of other transportation modes, etc., as shown in Figure 10b.

### 4.2.4. Guiding Sign Setting at Station Hall

The locations with high stopping density in the station hall of Xiaozhai station are found out according to the distribution of stopping points and duration recorded on the thumbnail. After analysis of the questionnaire, lack of information and attraction of facilities were found to be the main causes of pedestrian stopover. To relieve this problem, paths of pedestrians with different travel purposes should be divided according to the spatial characteristics of the station, so as to ensure that pedestrians can reach the destination as soon as possible without interference. Meanwhile, as the station hall is the central location of the station and provides service to all passengers, the effective setting of signs should be ensured. During the field investigation, it was found that pedestrians felt confused at the moment when entering the station hall through stairs or passageways. Therefore, signs should be set at the space connection to reduce unnecessary stay of pedestrians in the station hall. Guiding signs in station hall areas can be categorized as suspension, ground, wallpaper, and standing, etc. For locations with high pedestrian density, suspended guiding signs are recommended as an efficient means of pedestrian guidance, and coupled with grounded guiding signs as an information supplement. It is noteworthy that grounded guiding signs should not be set on the way of passenger flow or interweaving part, avoiding the disturbance of the passenger flow caused by stagnation of pedestrians. In addition, it is necessary to strengthen the setting before the diversion points to eliminate the conflict points caused by the return of pedestrians.

### 4.2.5. Guiding Sign Setting at Platform

The platform serves pedestrians of boarding and alighting. Passengers' information needs are mainly for outbound guidance and transfer guidance. Sufficient and eye-catching signs should be provided due to the large passenger flow of people. The contents of guiding signs mainly include the information of the direction, station name, exit guidance, stair guidance, elevator guidance, and time. At the platform area, information such as ride, transfer, and exit are recommended to use suspended and grounded signs, as shown in Figure 11a. A tube map should be set in the middle and both ends of the platform to facilitate pedestrians to travel (see in Figure 11b). Also, composite settings of guiding signs with suspension type and standing type (see in Figure 11c), and suspension type and ground type (see in Figure 11d) can also be used.

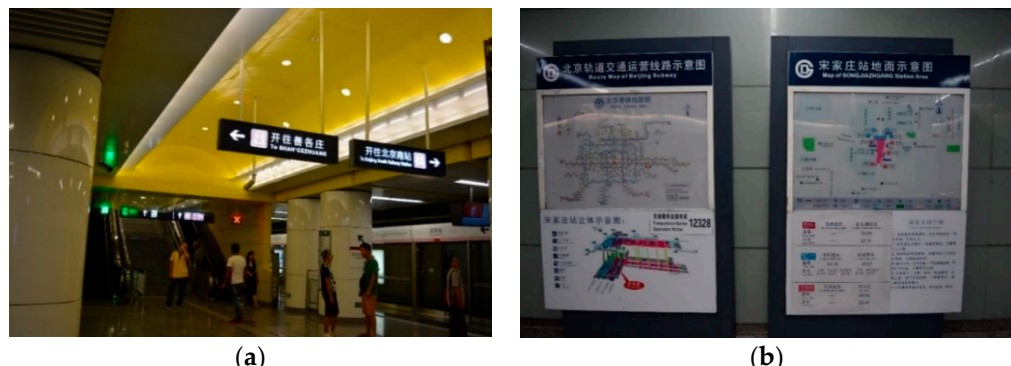

(**a**)                                                     (**b**)

**Figure 11.** *Cont.*

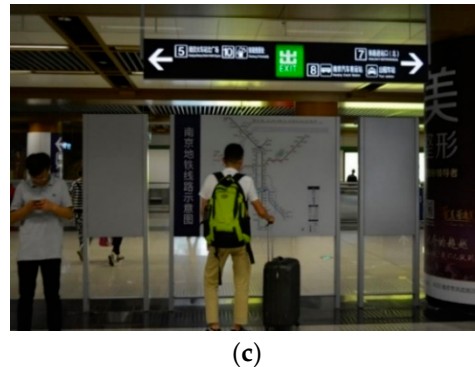
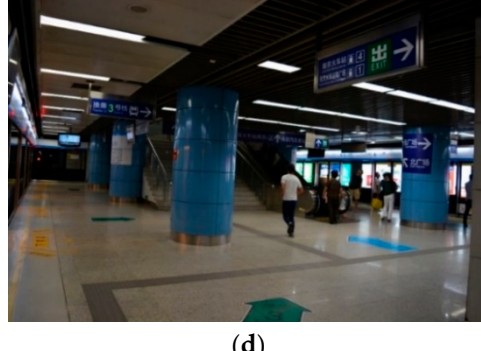

(**c**)                                                                                          (**d**)

**Figure 11.** Different types of guiding signs: (**a**) Suspension type; (**b**) wallpaper; (**c**) suspension and standing type; (**d**) suspension and grounding type.

### 4.3. Verification of Proposed Guiding Sign Settins

After resetting of the guiding signs in Xiaozhai metro station according to the setting method proposed in Section 4.2, the viability and effectiveness of the proposed setting method needs to be verified. Normally, two general ways to acquire data are favored by most of researchers, one is observation and survey method, and the other is test method [26]. The data of walking time acquired by the test method often contains the subjective factors which might not be adapted for the practical situation of metro station. Therefore, average walking time of passenger flow at different locations in the station were recorded with assistance of the video observation method [17]. In assurance of similar passenger flow for analysis the effectiveness of adjusted guiding sign setting methods, the same time of a weekday was selected for recording before and after the adjustment of guiding sign.

The comparison of average walking time of passenger flow before and after resetting of guiding signs at different parts of the metro station is shown in Table 2.

**Table 2.** Comparison of average walking speed at different parts of metro stations before and after adjustment.

| Origin | Destination | Platform | Stairs (s) | Station Hall (s) | Sum (Before) (s) | Sum (After) (s) | Improvement (s) | PERCENTAGE |
|--------|-------------|----------|------------|------------------|------------------|-----------------|-----------------|------------|
| line 3 | exit AD | 20.11 | 45.00 | 110.78 | 175.89 | 151.67 | 24.22 | 13.8% |
| line 3 | exit BC | 30.33 | 47.17 | 96.17 | 173.67 | 154.34 | 19.33 | 11.1% |
| line 3 | exit EF | 27.80 | 51.80 | 48.00 | 127.60 | 116.00 | 11.60 | 9.1% |
| line 2 | exit AD | 20.11 | 30.67 | 19.78 | 70.56 | 64.12 | 6.44 | 9.1% |
| line 2 | exit BC | 25.11 | 26.00 | 20.12 | 71.23 | 67.43 | 3.80 | 5.3% |
| line 2 | exit EF | 19.20 | 28.00 | 83.00 | 130.20 | 116.70 | 13.50 | 10.3% |
| line 2 | line 3 | 16.16 | 19.91 | 30.91 | 66.98 | 62.01 | 4.97 | 7.4% |

As can be seen from Table 2, evident improvement has been observed after adjustment of guiding signs according to the method proposed in Section 4.2. The average walking time of different origin-destination (OD) pairs experienced an improvement from 5.3% to 13.8%. It is noteworthy that the walking time improvement of passenger flow from line two and line three to exit EF is of great importance as it linked the metro station and one of the busiest malls of Xi'an city. The passenger flow is great and minor improvement would be meaningful to social economy.

## 5. Conclusions

Guiding signs are of great importance on pedestrian guidance in the metro station. Currently, most studies on guiding signs focus on architecture, aesthetics and simulation. However, perspectives from humanization of pedestrian guidance signs such as pedestrian behavior needs and pedestrian cognition were seldom proposed. In this paper, the microscopic behavior characteristics data of pedestrians at different positions in typical metro stations were collected through pedestrian tracking experiments. By analysis of obtained data, guiding signs setting method at different locations

were studied concerning the pedestrian information demand and observed microscopic behavior characteristics. The research results of this paper can provide theoretical support and technical guidance for the optimal establishment of pedestrian guiding signs in metro stations with massive passenger flow. The main findings of the study are as follows:

(1) The walking speed of pedestrians in different types of passageways is different. The enclosed passageway has the fastest walking speed, followed by the open passageway and the semi-enclosed passageway.

(2) The walking speed of pedestrians at the stairs adjacent to the platform is higher than that not adjacent to the platform.

(3) The walking time showed a trend of decreasing first and then increasing with crowd density.

After resetting of the guiding signs at different parts of Xiaozhai station, evident improvements have been observed for different OD pairs. Due to the diverse forms of metro stations and the complex characteristics of pedestrians, the research methods of guiding sign setting optimization are also multiple. The limitation of this study is that the proposed resetting guiding sign methods were only tested in Xiaozhai station because of multiple practical difficulties; the viability still needs to be verified at other stations in future to organize a general guiding sign setting guide. If the relationship between architectural features and pedestrian behavior characteristics can be quantified, the guiding signs can be better optimized to improve environment and efficiency of passenger flow.

**Author Contributions:** Conceptualization, J.X.; methodology, B.L.; software, M.L.; validation, M.L., B.L. and J.L.; formal analysis, B.L.; investigation, H.L. and Z.C.; resources, Y.H.; data curation, H.L. and Y.Z.; writing—original draft preparation, B.L.; writing—review and editing, J.X. and M.L.; supervision, J.X.; project administration, J.X.

**Funding:** This research was funded by the Scientific Research Project of Transportation Department of Shaanxi Province (16–40 K).

**Acknowledgments:** The opportunity to explore this topic was made possible by funding provided by the Scientific Research Project of Transportation Department of Shaanxi Province and China Harbor Engineering Company Limited.

**Conflicts of Interest:** The authors declare no conflict of interest.

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
