# Peer review of "Enhancing Role of Guiding Signs Setting in Metro Stations with Incorporation of Microscopic Behavior of Pedestrians"

_sustainability, doi:10.3390/su11216109_

Round 1

Reviewer 1 Report

No comments.

Author Response

Since you had not commented on the manuscript, therefore, the manuscript was mainly rectified by comments of academic editor and reviewer 2 and reviewer 3. Please check the revised version of manuscript 610138.

Reviewer 2 Report

The paper addresses a quite interesting research question. The proposed methodology is convincing. The results and discussion are adequate. The conclusions seem properly supported

The text is understandable but full of minor grammatical errors. The paper should be proof-read by a native English speaker.

I would have liked to see the authors relate their findings to other literature

It will be helpful to understand why the project is necessary and current literature and related policies

Author Response

The paper addresses a quite interesting research question. The proposed methodology is convincing. The results and discussion are adequate. The conclusions seem properly supported

Point 1: The text is understandable but full of minor grammatical errors. The paper should be proof-read by a native English speaker.

Response: Thanks for your precious comment, the text has been checked by a native English speaker to improve the language level of this paper, please check it in the revised version of manuscript 610138.

Point 2: I would have liked to see the authors relate their findings to other literature

Response: Thanks for your comment on this part. It is helpful to improve the quality of the introduction part of this paper. The manuscript has been rectified according to your comments. Please check it in the revised version of manuscript 610138.

Point 3: It will be helpful to understand why the project is necessary and current literature and related policies

Response: Thanks for your precious comment. The manuscript has been rectified according to your comments. The innovative contribution and necessity of this study also highlighted in the revised manuscript. The study might fill the gap that lacking of relevant study focused on improvement of efficiency of pedestrian flow in metro station under normal situation from perspective of pedestrian psychology and microscopic behavior. Moreover, the research results of this paper can provide theoretical support and technical guidance for the optimal establishment of pedestrian guiding signs in metro stations with massive passenger flow.

Please check it in the revised version of manuscript 610138.

Reviewer 3 Report

The paper studies the behaviour of passengers in a subway station. In my opinion, the article cannot be published because it has many weaknesses.

Firstly, the title is incorrect. The observation of users is not itself a way to improve the efficiency of driving signals in a station, but a method that could provide the tools to achieve it.

More importantly, the results obtained do not refer to such a possible improvement. In fact, to get such a result, it would be necessary to change the signals and test the behaviour of the users again, to see if the new system is more efficient than the previous one. This is also confirmed by the results reported on lines 365-370 which refer only to passenger behaviour, without any reference to signal improvement.

The proposed study, then, only on a subway station, cannot be easily generalized. For example, as also mentioned by the authors, occasional users, who do not know the station, behave differently from commuters, who know it well. This can lead to entirely different results between stations affected by tourist traffic and others, where the percentages of different types of users are entirely different.

The literature should be improved a lot. Even the chapter on pedestrians in the Highway Capacity Manual is not quoted.

Minor
Line 128 "microcosmic" should be "microscopic".

Author Response

Point 1: Firstly, the title is incorrect. The observation of users is not itself a way to improve the efficiency of driving signals in a station, but a method that could provide the tools to achieve it.

Response: Thanks for your precious comment, the title has been changed to “Enhancing Role of Guiding Signs Setting in Metro Stations with Incorporation of Microscopic Behavior of Pedestrians”. Please check the revised version of manuscript 610138.

Point 2: More importantly, the results obtained do not refer to such a possible improvement. In fact, to get such a result, it would be necessary to change the signals and test the behaviour of the users again, to see if the new system is more efficient than the previous one. This is also confirmed by the results reported on lines 365-370 which refer only to passenger behaviour, without any reference to signal improvement.

Response: Thanks for your precious comment. Section 4.3 has been added to the revised version of manuscript as a verification of the improvement of the proposed guiding sign setting method. In assurance of similar passenger flow for analysis the effectiveness of adjusted guiding sign setting methods, the same time of a weekday was selected for recording before and after the adjustment of guiding sign. The comparison of average walking time of passenger flow before and after resetting of guiding sign at different parts of the metro station is shown in the revised manuscript. Please check the revised version of manuscript 610138.

Point 3: The proposed study, then, only on a subway station, cannot be easily generalized. For example, as also mentioned by the authors, occasional users, who do not know the station, behave differently from commuters, who know it well. This can lead to entirely different results between stations affected by tourist traffic and others, where the percentages of different types of users are entirely different.

Response: Thanks for your precious comment. We are agree with your perspective that “The proposed study, then, only on a subway station, cannot be easily generalized”. It is also an limitation of this study, therefore, to minimize its influence on the result, we choose aggregate parameters to verify its effectiveness. Here, average walking time was used as the index for verification, the data were obtained with assistance of the video observation method. Please check the revised version of manuscript 610138.

Point 4: The literature should be improved a lot. Even the chapter on pedestrians in the Highway Capacity Manual is not quoted.

Response: Thanks for your precious comment. The literature part in section one has been reorganized and considerable number of papers have been added to the manuscript to improve the quality of the introduction, also works of recent years has also been added. The innovative contribution and necessity of this study also highlighted in the revised manuscript. Please check it in the revised version of manuscript 610138.

Point 5: Minor Line 128 "microcosmic" should be "microscopic".

Response: Thanks for your precious comment, the linguistic mistake has been rectified, please check it in the revised version of manuscript 610138.

Round 2

Reviewer 3 Report

Authors have improved sufficiently the paper.